# Gender-specific hearing loss in German adults aged 18 to 84 years compared to US-American and current European studies

Petra von Gablenz[1]*, Eckhard Hoffmann[2], Inga Holube[1]

1 Institute of Hearing Technology and Audiology, Jade University of Applied Sciences and Cluster of Excellence "Hearing4all", Oldenburg, Germany, 2 Department of Audiology, Aalen University of Applied Sciences, Aalen, Germany

* petra.vongablenz@jade-hs.de

**Data Availability Statement:** All relevant data are contained within the manuscript. Additionally, data

## Abstract

### Introduction

From an epidemiological point of view, the increase of pure-tone hearing thresholds as one aspect of biological ageing is moderated by societal factors. Since health policies refer to empirical findings, it is reasonable to replicate population-based hearing surveys and to compare estimates for different birth cohorts from the same regions or, conversely, for the same birth cohorts from different regions.

### Methods

We pooled data from two independent cross-sectional German studies conducted between 2008 and 2012 and including 3105 adults. The increase of thresholds, the prevalence and risk of hearing impairment (HI) by age and gender were compared to results reported for European and US-American studies that were carried out at about the same time. Since these studies differed with regard to the age limits, the statistical approaches and, importantly, their definitions of HI, data adjustments were performed to enable the comparison.

### Results

Overall, 15.5% of the participants in the German studies showed a pure-tone average at 0.5, 1, 2, and 4 kHz in the better ear (PTA) greater than 25 dB HL and 8.6% had a PTA of at least 35 dB HL. Based on one-to-one comparisons, the German estimates demonstrated a good agreement to a large Dutch study and with some reservations to a Swedish study, but considerable differences to US-American results. Comprehensive comparisons of the within-study gender differences showed that age-related HI was less and the gender gap was markedly smaller in Europe compared to the US due to the lower HI in males found in the European studies.

is available from http://dx.doi.org/10.5061/dryad. cfxpnvx27 without any restriction.

**Funding:** Publication was supported by the German Research Foundation (DFG) and the Open Access Publication Funds of Jade University of Applied Sciences. Data collection in HÖRSTAT was funded by the Lower Saxony Department of Science and Culture and the European Regional Funding (EFRE). Further analysis was financed from the research fund of the Jade University of Applied Sciences and the federal resources of Niedersächsisches Vorab by the Ministry of Science and Culture of Lower Saxony within the research focus "Hearing in everyday life (HALLO)." Auritec Medizindiagnostische Systeme GmbH (www.auritec.de) provided portable Ear 2.0 audiometers (ih, pvg). The Federal Armed Forces financed the study "Hearing in Germany" and gave further support by providing a mobile examination unit from WTD91 in Meppen, Germany (eh).

**Competing interests:** The authors have declared that no competing interests exist.

## Conclusion

Discrepancies in measurement procedures, conditions, and equipment that complicate the comparison of *absolute* HI estimates across studies play no or only a marginal role when comparing *relative* estimates. Hence, the gender gap differences reviewed in this analysis possibly stem from societal conditions that distinguish societies commonly labeled modern industrialized western countries.

## Introduction

Age-related hearing loss is a gradual process that advances almost unnoticed from an individual perspective, but has long been recognized as a major health issue in ageing societies. Internationally, a considerable number of studies has examined pure-tone hearing in population-based samples. Since most of these studies followed a cross-sectional design, they provided snapshots of hearing thresholds, pure-tone averages or prevalence figures of hearing impairment, primarily as a function of age and gender. On this note, the present contribution expands the international database on age- and gender-related hearing loss and reviews the main findings derived in two cross-sectional German studies when compared to recent international results.

Living and working conditions, health care and life-style factors, which in general affect how people age and impact hearing in particular, are critically important variables. Thus, almost needless to say, they change with time and vary according to locality within and between societies. This is one of the main reasons why international standards such as ISO 7029 [1] and ISO 1999 [2], which rely on epidemiological data, need to be periodically updated. ISO 7029 describes the statistical distribution of hearing threshold deviation by age and gender. ISO 1999 estimates the impact of noise exposure and refers to ISO 7029. After ISO 7029 was revised in 2017, ISO 1999 is, strictly speaking, also due for revision—all the more since various recent studies indicated that the process of age-related hearing loss is slowing down in modern western societies.

Zhan et al. found generational differences when comparing the prevalence of hearing impairment (HI) in adults aged 45 to 94 years between four birth cohorts recruited for the Epidemiology of Hearing Loss Study (EHLS) and the Beaver Dam Offspring Study, Beaver Dam, Wisconsin [3]. This effect was stronger in men than in women. Since the age-specific prevalence was lower in more recent birth cohorts for age-bands below 85 years, the authors concluded that older adults may be retaining good hearing longer than previous generations. In the National Health and Nutrition Examination Survey (NHANES) conducted between 1999 and 2004, Hoffman et al. found better age-specific hearing threshold levels (HTL) in 25 to 64 year-old men and women, particularly at high frequencies, than in a comparable survey conducted between 1959 and 1962 [4]. Similarly, the comparison of NHANES data collected from 1999 to 2004 and 2011 to 2012 showed a decline of age-adjusted prevalence of HI for both genders [5].

This trend towards lower HTL and HI in US-American surveys was also observed in current European data. Based on the data sampled in the Gerontological and Geriatric Population Study in Gothenburg, a longitudinal study on the health of 70 year-old people, Hoff et al. compared HTL in Swedish 70-year-olds over four decades and found substantial improvements over time [6]. Rosenhall, Möller & Hederstierna, however, referring to data from the same

study in Gothenburg, stated that the hearing of 75-year olds had not changed over three decades [7].

Data of the German study HÖRSTAT, which form part of the present analysis, was derived in adults aged from 18 to 97 years in the years 2010 to 2012. Since no appropriate national surveys were available for comparison, the main US-American and earlier European surveys were used to compare results. HTL and prevalence of HI in men, not in women, were found to be considerably lower than reported in those international studies, leading to comparatively small gender differences in the German study [8,9]. About the same time (2011 to 2015), pure-tone hearing of adults aged 50 years and older was tested in the course of the large prospective Rotterdam study that addressed health in the elderly [10]. Homans and collegues contrasted HTL and the pure-tone average at 0.5, 1, 2, and 4 kHz (PTA) to results from US-American studies that were carried out 20 to 30 years earlier, the Framingham study [11] and the Beaver Dam study [12]. Depending on age group and frequency, they found overall better hearing thresholds in Dutch men than in American men of the same age. In women, however, basically the reverse was the case. As in the German study, the gender differences were smaller than in the comparison studies. Homans et al. applied the Global Burden of Disease (GBD) criterion for disabling hearing loss [10], originally proposed by Stevens et al. [13], arguing that moderate HI with PTA$\geq$35 dB defines "the level at which intervention is definitively beneficial". Above the age of 65, the prevalence of $HI_{GBD}$ in the better ear was 33% in men and 31% in women in the population-weighted sample. The authors assumed that this prevalence was representative not only for the Dutch population, but likely for most populations in Western Europe.

Against this background, we pooled and reanalyzed data from two independent German studies to examine whether this assumption of Homans et al. holds for the middle-aged and elderly population in Germany. Giving special attention to gender differences, HTL and PTA increases with age for the pooled sample were exemplarily compared to results from the most recent population-based surveys in Europe, the above-mentioned Dutch and Swedish studies, and to the US. Additionally, the hearing-aid adoption rate is reported separately by age and gender. The German data was collected in the studies "Hearing in Germany" and HÖRSTAT, which were conducted roughly at the same time in geographically distant places by different research groups. The study protocols shared e.g., pure-tone audiometry as well as various questionnaire items. These databases were already used to extrapolate the prevalence and degree of hearing impairment in Germany in population projections taking the medium-term demographic trend into account [14].

Unfortunately, the comparison international surveys differ with respect to the distribution and categorization of age, of HTL measurement frequencies, of definitions of HI prevalence, and in their descriptive statistics. In order to compare differences of HTL, HI prevalence and the respective gender gaps, our data was reanalyzed, and one-to-one aligned to the descriptive parameters and statistics used in the international studies, including adjustments of the study sample's age distribution. So far, no HTL data have been reported for the NHANES cycles 2011 to 2012. Therefore, we also included prevalence and risk estimates for HI from the 2011–2012 cycles [5], in order to retain the comparison to the narrow time frame of data collection from 2008 to 2015.

Our hypotheses build on the reasoning that "environmental, lifestyle, or other modifiable factors might contribute to the etiology of hearing impairment in older adults" [3]. In concrete terms, we assumed that the findings in the German samples reflect the decrease of age-related hearing loss noted for other western countries and, particularly, that the European studies yield similar results owing, e.g., to cultural similarities and social commonalities consolidated over decades in the progressive political and economic integration of European societies.

## Materials and methods

### Studies and subjects

HÖRSTAT was carried out by the Institute of Hearing Technology and Audiology in Oldenburg between 2010 and 2012. The survey included a total of 1903 adults from Oldenburg and Emden. These cities differ most notably in their economic structure. Thus, stratified samples of balanced size were calculated to approximate the national distribution of age, gender, and employment in the industrial sector. Participants were randomly selected from the local registration offices and initially contacted by letter mail and a reminder, followed by personal and telephone contact in case of failure to respond. Thirty-eight percent of the participants passed all stages of this recruiting procedure. The response rate was 21% in total and 30% on average in the age group from 40 to 80. HÖRSTAT included a non-responder interview to control for possible response biases. Non-responders and participants, both were queried regarding hearing difficulties. And besides, participants can be divided into early responders (spontaneous response after invitation letters) versus late responders (response after additional telephone call) and compared with regard to hearing difficulties and PTA. The comparison of responders and non-responders as well as the comparison of early and late responders did not indicate systematic or larger between-group differences. The design and conduct of the survey, including non-response and late response analysis, were described in detail elsewhere [8]. For the present analysis, the data of 1866 adults with complete and valid audiometric measurements at the octave frequencies between 0.5–4 kHz were used, 585 participants from Emden (324 female) and 1281 participants from Oldenburg (691 female). Data of 37 adults were excluded because the measurements were compromised by various factors.

The study "Hearing in Germany" was conducted in Aalen, a middle-sized town in southwest Germany. The survey of the University of Applied Sciences Aalen was split into three waves. The first and the third waves in May/June 2008 and May 2009, respectively, were based on a random sample of 15,000 subjects aged 7 years and older, derived from the Aalen registration office. For each 5-year age band, an equal number of addresses of males and females were drawn. Due to shortcomings in the study administration, the response rate can only be specified for the first wave and was 13%, mainly due to the extremely low participation of young adults. The response rate increased from the middle-age groups on to retirement age and reached a peak of 25% in the 65-year old cohorts of both genders. The execution and results of this study were summarized in an internal report and partially published [15]. For the present analysis, complete data of air-conduction thresholds between 0.5 and 4 kHz from both ears were available for 1239 adults (645 females).

The research design and procedures of HÖRSTAT and the Aalen study were in accordance with the principles of the Declaration of Helsinki and written informed consent was obtained from all participants. The HÖRSTAT study was approved by both by the data protection officer of the Jade University of Applied Sciences and the ethics commission of the Carl von Ossietzky University in Oldenburg. The Aalen study was approved by the Clinical Research Organization / Freiburger Ethikkommission International.

To check for substantial differences in calibration, the audiometric data derived in 18 to 25 year-old participants from HÖRSTAT and the Aalen study were analyzed. Major differences in hearing thresholds were neither expected nor could be tolerated for these very young adults. As the grouped median hearing thresholds at 0.5, 1, 2, and 4 kHz differ by only 0.6 on average (1.2 dB maximum) in this age group, there is no evidence for calibration differences that need to be taken into account when pooling.

In total, merging the study data from the HÖRSTAT and the Aalen study resulted in 3105 records. Participants in both studies did not receive any monetary compensation.

## Pure-tone audiometry

HTL via air conduction were measured at the frequencies 0.25, 0.5, 1, 2, 3, 4, 6, and 8 kHz, separately in the left and the right ear. Following an otoscopic examination, measurements were performed manually according to ISO 8253–1:2010 [16] using a 5-dB step size by trained personnel who were mostly professional hearing-aid acousticians, but also by advanced students from the universities' audiology study courses. If the thresholds exceeded the audiometer range, for this analysis they were set at the maximum presentation level of the audiometer plus 5 dB.

In HÖRSTAT, testing was carried out during home visits (8%) or in the institute's own quiet facilities (92%). Ambient noise levels were monitored by measuring the noise level in 1/3-octave bands, to ensure that requirements of ISO 8253–1 were met [17]. Threshold measurements were conducted using ten Auritec mobile audiometers ear2.0 and Sennheiser HDA200 headphones. All equipment was new and calibrated by the manufacturer, complying with IEC 60645–1 [18] and ISO 389–8 [19] at the start and in the midterm of the study. In the Aalen study, all hearing tests were performed in sound-attenuation booths using four Maico audiometers MA 55 and HDA200 headphones, maintained and calibrated annually according to current standards.

## Parameters and statistics

HTL data in both gender groups showed the so-called better-ear effect, with poorer HTL at 3, 4, and 6 kHz in left ears than in right ears and better HTL in left ears at 1 kHz than in right ears. Left and right ears differed by 1–2 dB on average. An ANOVA was performed in an all-ears data set with laterality as the independent variable and the subject number as covariate, although the conditions for parametric statistics were not fully met. The left-right ear differences reached significance with p-values ranging between .01 and .05 in the total sample, but were not confirmed in post-hoc tests for each age group by gender. The small scale and instability of the left-right ear difference justified an averaging of the right and left ear HTL, in order to estimate the 10th, 25th, 50th, 75th, and 90th percentile at the frequencies 0.25, 0.5, 1, 2, 3, 4, 6, and 8 kHz. Note that the data for this article are available from the Dryad Repository (http://dx.doi.org/10.5061/dryad.cfxpnvx27). HTL data at either 3, 6, or 8 kHz were randomly missing in less than 2% of the cases. Generally, measurement data were regarded as grouped data using the algorithm implemented in SPSS 25.0 [20]. As outlined by Dobie [21] the grouped data approach is less sensitive to small distributional changes, because it assumes an even distribution of the 'real' HTL within the 5-dB step size used in audiometry. PTA was calculated based on the averaged HTL of the right and left ear. The prevalence of hearing impairment is reported for the WHO criterion (better ear PTA > 25 dB HL) and the competing GBD scheme (PTA $\geq$ 35 dB HL), hereinafter referred to as $HI_{WHO}$ and $HI_{GBD}$, respectively. Results are given separately for men and women in 10-year age groups centered on 30, 40 years etc., with the exception of the youngest 20 years group, that comprises participants aged 18 to 24 years. Data for adults aged 85 years and older (n = 60) were scarce and not displayed in Tables and Figures, but included for estimating the overall HI prevalence. 95% confidence intervals (CI) are given for all prevalence estimates and the hearing-aid adoption rate to allow for evaluating between-group differences. Additionally, Pearson $Chi^2$ test was performed (2-sided, significance level 0.05).

Higher levels of school and professional qualifications were disproportionately represented both in HÖRSTAT [22] and in the Aalen survey [14]. Weights were applied to adjust the distribution of vocational levels, separately for age and gender, to population statistics for the year 2011 when HTL and HI prevalence were reported separately for age and gender [23]. All

results for the total sample (N = 3105), including participants aged 18 to 97 years, were estimated with weights applied for age, gender, and educational status to approximate the national distribution for the year 2011 [23,24]. The number of participants in each gender and age group and the population-weighted percent share are given in Table 1. Gender-specific estimates were adjusted for age assuming equal gender proportions in the population. These weighting models were considered highly acceptable, because 94 to 99% of the weighting factors ω met the condition $0.3 < \omega < 3$. Critical factors with $\omega \geq 3$ account for maximally 1% and refer to very young or very old age groups (below 30, 85 and older).

## Adjustments for comparison to international results

Results derived in the pooled sample were compared to findings reported for the Dutch Rotterdam study [10], the most recent results for the Swedish Gothenburg study [6], the US-American NHANES cycles 1999–2004 [25,4], the cycles 1999 to 2006 [26], and the cycles 2011 to 2012 [5]. Although principally eligible for comparison, the HTL estimates for an earlier Gothenburg cohort were not included in this comparison because the authors stated HTL deviations to earlier cohorts that possibly originated from "differences regarding ambient noise levels in the test rooms used, in combination with difficulties for less experienced testers to detect the problems that arise close to 0 dB HL in a non-insulated test room"[7].

**Comparing PTA increase by age.** In the case of the US-American NHANES data from the cycles 1999 to 2006, Hoffman et al. reported interpolated HTL medians estimated for the better ear on a frequency-by-frequency basis [4,26]. We replicated this approach in our data set. PTA based on the averaged HTL of the right and left ear was on average 2.6 dB higher (worse) than the PTA based on better-ear-by-frequency HTL. This difference induced by the method increased linearly with PTA. Assuming an overall similar association in the original NHANES data as in our data set, the PTA estimated from the HTL values reported by Hoffman et al. were adjusted for graphical display using the following Formula:

$$PTA_{NHANES\ adjusted} = 1.1 \cdot PTA_{NHANES} + 1.8 \tag{1}$$

Homans et al., who reported mean HTL and standard error of measurement for the Rotterdam study [10], kindly provided conventionally estimated HTL and PTA percentiles for their large data set. Estimates using the midpoint or grouped-data approach yield slightly different percentiles than the conventional method. Since the absolute differences are overall non-systematic and mostly range between ± 1 dB, they were neglected for the purpose of a graphical comparison of PTA increase by age. For the Swedish study, Hoff et al. reported interpolated percentiles, thus, no adjustments were made [6].

To obtain comparison estimates methodologically as close as possible, PTA was calculated from age- and gender-specific median HTL for all studies. If results were reported separately for left and right ears, the median estimates were averaged before PTA calculation. PTA increase by age was approximated using a quadratic function. These approximations were almost perfect, with $R^2 = .99$. Additional weighting to consider uncertainty of the curve estimation using, e.g., group sample size and interquartile range, was waived, because it showed only minimal effects that were indistinguishable in the plots. Gender differences in HTL, PTA, and HI prevalence were calculated simply by subtracting values for females from corresponding values for males.

**Comparison to US data, 2011 to 2012.** The NHANES cycle 2011 to 2012 would give an almost perfect match with respect to the data collection period, but the HTL distribution was not published for this cycle. Hence, the corresponding prevalence and risk estimates for adults aged 20 to 69 years as reported by Hoffman et al. [5] for the most current NHANES data were

**Table 1. Distribution of HTL averaged across left and right ears for females and males.**

| | | Female | | | | | | | Male | | | | | | |
|---|---|---|---|---|---|---|---|---|---|---|---|---|---|---|---|
| Age (years) | | 20 | 30 | 40 | 50 | 60 | 70 | 80 | 20 | 30 | 40 | 50 | 60 | 70 | 80 |
| n = | | 114 | 159 | 208 | 365 | 335 | 262 | 198 | 75 | 119 | 213 | 229 | 305 | 244 | 220 |
| % weight[a] | | 4.7 | 7.1 | 8.0 | 9.8 | 7.8 | 7.0 | 5.0 | 4.9 | 7.3 | 8.1 | 10 | 7.5 | 6.3 | 3.5 |
| Frequency | Percentile | | | | | | | | | | | | | | |
| 0.25 kHz | 10 | -3 | -3 | -2 | -1 | 1 | 4 | 7 | -2 | -3 | -2 | -2 | 1 | 3 | 6 |
| | 25 | 0 | -1 | 0 | 2 | 4 | 7 | 11 | 0 | -1 | 0 | 1 | 4 | 6 | 10 |
| | Median | 3 | 3 | 4 | 5 | 8 | 12 | 17 | 2 | 2 | 3 | 5 | 8 | 12 | 16 |
| | 75 | 7 | 6 | 8 | 10 | 14 | 21 | 27 | 6 | 6 | 7 | 10 | 14 | 19 | 25 |
| | 90 | 10 | 10 | 13 | 16 | 20 | 33 | 37 | 11 | 10 | 11 | 16 | 24 | 29 | 37 |
| 0.5 kHz | 10 | -3 | -3 | -2 | -1 | 2 | 5 | 8 | -2 | -4 | -3 | -1 | 1 | 3 | 7 |
| | 25 | 0 | -1 | 1 | 2 | 5 | 9 | 13 | 0 | -1 | 0 | 2 | 5 | 7 | 11 |
| | Median | 3 | 2 | 4 | 6 | 9 | 15 | 19 | 2 | 2 | 3 | 6 | 10 | 14 | 18 |
| | 75 | 7 | 6 | 9 | 11 | 15 | 23 | 32 | 6 | 6 | 8 | 10 | 16 | 22 | 29 |
| | 90 | 11 | 11 | 16 | 18 | 23 | 35 | 45 | 10 | 12 | 12 | 17 | 25 | 32 | 41 |
| 1 kHz | 10 | -4 | -2 | -1 | 1 | 2 | 5 | 8 | -4 | -4 | -2 | -1 | 2 | 5 | 9 |
| | 25 | -1 | 0 | 1 | 3 | 5 | 9 | 14 | -1 | -1 | 1 | 2 | 6 | 9 | 14 |
| | Median | 1 | 3 | 4 | 7 | 9 | 16 | 24 | 2 | 2 | 4 | 6 | 11 | 16 | 23 |
| | 75 | 5 | 7 | 9 | 12 | 15 | 26 | 37 | 5 | 5 | 8 | 11 | 17 | 24 | 34 |
| | 90 | 9 | 12 | 16 | 19 | 24 | 38 | 49 | 10 | 13 | 14 | 18 | 26 | 36 | 48 |
| 2 kHz | 10 | -2 | -2 | 0 | 2 | 5 | 9 | 14 | -3 | -3 | -2 | 0 | 4 | 10 | 17 |
| | 25 | 1 | 1 | 3 | 5 | 9 | 14 | 24 | 0 | -1 | 1 | 4 | 9 | 15 | 26 |
| | Median | 4 | 5 | 7 | 10 | 14 | 23 | 35 | 3 | 3 | 5 | 8 | 15 | 25 | 39 |
| | 75 | 7 | 9 | 11 | 17 | 22 | 36 | 48 | 7 | 8 | 10 | 15 | 26 | 36 | 52 |
| | 90 | 12 | 13 | 19 | 23 | 34 | 46 | 60 | 12 | 14 | 19 | 29 | 36 | 50 | 63 |
| 3 kHz | 10 | -5 | -3 | -1 | 1 | 5 | 11 | 16 | -5 | -2 | -2 | 3 | 8 | 16 | 24 |
| | 25 | -2 | -1 | 1 | 5 | 9 | 17 | 26 | -1 | 1 | 2 | 6 | 15 | 27 | 41 |
| | Median | 1 | 3 | 5 | 9 | 16 | 25 | 43 | 3 | 4 | 7 | 13 | 24 | 40 | 54 |
| | 75 | 5 | 7 | 10 | 17 | 25 | 40 | 53 | 7 | 8 | 16 | 25 | 39 | 53 | 64 |
| | 90 | 8 | 12 | 17 | 26 | 38 | 52 | 59 | 12 | 15 | 29 | 42 | 53 | 63 | 77 |
| 4 kHz | 10 | -6 | -4 | -2 | 1 | 5 | 13 | 18 | -5 | -2 | -1 | 6 | 13 | 22 | 32 |
| | 25 | -3 | -2 | 1 | 5 | 11 | 20 | 28 | -2 | 0 | 3 | 10 | 22 | 34 | 50 |
| | Median | 0 | 2 | 5 | 10 | 17 | 31 | 47 | 2 | 4 | 9 | 17 | 33 | 47 | 60 |
| | 75 | 4 | 6 | 9 | 19 | 27 | 44 | 57 | 6 | 9 | 22 | 30 | 47 | 59 | 69 |
| | 90 | 7 | 13 | 18 | 27 | 43 | 56 | 66 | 10 | 15 | 33 | 46 | 61 | 68 | 82 |
| 6 kHz | 10 | -4 | -1 | 2 | 5 | 12 | 19 | 29 | -4 | -2 | 1 | 8 | 16 | 29 | 43 |
| | 25 | -1 | 2 | 6 | 9 | 17 | 27 | 41 | 1 | 2 | 6 | 13 | 26 | 40 | 58 |
| | Median | 4 | 5 | 12 | 16 | 26 | 41 | 54 | 5 | 8 | 13 | 22 | 40 | 54 | 67 |
| | 75 | 9 | 11 | 19 | 24 | 37 | 55 | 67 | 10 | 14 | 21 | 35 | 53 | 66 | 78 |
| | 90 | 14 | 17 | 26 | 37 | 51 | 66 | 78 | 15 | 20 | 37 | 49 | 63 | 76 | 87 |
| 8 kHz | 10 | -1 | 1 | 3 | 8 | 15 | 26 | 38 | -1 | 1 | 5 | 10 | 20 | 37 | 56 |
| | 25 | 3 | 6 | 7 | 12 | 22 | 37 | 59 | 2 | 5 | 10 | 16 | 28 | 49 | 67 |
| | Median | 7 | 10 | 14 | 19 | 33 | 55 | 69 | 6 | 10 | 16 | 26 | 47 | 64 | 77 |
| | 75 | 12 | 15 | 21 | 29 | 47 | 69 | 78 | 12 | 16 | 26 | 41 | 60 | 75 | 88 |
| | 90 | 19 | 20 | 29 | 42 | 62 | 78 | 92 | 20 | 22 | 43 | 57 | 74 | 82 | 97 |

Weights applied to adjust educational levels to population statistics. Age is the midpoint of age bands 25 to 34, 35 to 44 etc. except the youngest band that refers to 18 to 24 yr adults.

[a] Population-weighted percent adults aged 18 and more years. Age groups 85 years and older account for 2.1% (females) and 0.8% (males).

additionally consulted, to critically update the comparison to US surveys. Odds Ratios (OR) from logistic regression for the prevalence of $HI_{WHO}$ and high frequency hearing loss $HI_{high-Freq}$ with $PTA_{high}$ (3, 4, and 6 kHz) greater than 25 dB HL in the better ear were estimated with weights to adjust the age distribution in our sample to the US-American sample assuming equal gender shares.

## Results

### HTL distribution

The distribution of HTL estimated for gender and age subgroups in our data is given in Table 1, in which the percentile values were rounded to the next integer. Hearing sensitivity for females and males shows the commonly known pattern: HTL increase with age, particularly in the high frequencies and more strongly in males, and, at low frequencies, a somewhat stronger increase in females. Likewise, the typical distributional characteristics apply: Spread and positive skewness of HTL increase with age and with frequency, but stagnate and even reversed when a subgroup's HTL median of approximately 50 dB HL was reached.

Comparing the HTL distribution to the international results, considerable differences were noted. HTL in our sample were generally lower (better) than reported for the Dutch and the US populations, but overall higher (worse) than in the Swedish sample. With respect to the US data, these differences were small in females except for HTL at 0.5 kHz, but notable in males at 0.5 kHz and in the 3 to 6 kHz region, ranging mostly between 5 and 8 dB in the medians. In contrast, the differences of the age-specific mean HTL (left and right ear average, not shown here) between the Dutch and our estimates were smaller for males than for females, and basically limited to 4 kHz and frequencies below 2 kHz. In terms of dB, HTL differences at 0.25, 0.5, 1, and 4 kHz accounted for 3 to 6 dB in females but only 2 to 3 dB in males on average. Age-specific HTL differences at 2 and 8 kHz were $\leq$ 1 dB, thus well below the uncertainty due to equipment and calibration. Noticeable HTL differences to the Swedish data were observed, ranging mostly between 4 to 7 dB in the medians for males and about 2 to 3 dB in females.

### HI prevalence, asymmetric hearing, and hearing-aid use

The prevalence of $HI_{WHO}$ and $HI_{GBD}$, as well as asymmetric hearing and hearing-aid use, are summarized in Table 2 separately for gender and age groups. Additionally, $HI_{GBD}$ classes are shown in Fig 1 to specify the degree of hearing loss. Young- to middle-aged groups were collapsed both in the table and the graphical display because these HI criteria are not really distinctive in age groups below 50 years. The estimates' 95%-CI for females and males overlapped in all comparisons. Based on $CHI^2$ statistics, however, gender differences reached significance in the 60 and 80 years age groups as well as in the total sample for $HI_{WHO}$ with p < 0.01 and in the 80 years age group for hearing-aid adoption with p = 0.023. All other gender-related comparisons failed the significance criterion. Overall, the HI prevalence increase was steep in post-climacteric age groups, and more pronounced in males than in females. $HI_{GBD}$ prevalence, for instance, increased from about 1% in the collapsed 18–54 years groups to 15% in the 70 year- and 38% in the 80 year- age band. Overall, when sample weights were applied to achieve the distribution of age, gender, and professional education in the general German population of the year 2011, the $HI_{GBD}$ prevalence estimate is 8.6%. The hearing-aid adoption rate was 5.6% in the total sample, and lagged behind the $HI_{GBD}$ prevalence figures in every age and gender subgroup. The ratio of percent hearing-aid adoption rate and percent $HI_{GBD}$ was lowest in the 80 year subgroup. Interestingly, the hearing-aid adoption rate was 6.2% in adults with mild HI according the GBD classification (20 < PTA in the better ear < 35).

**Table 2. Prevalence of hearing loss based on the PTA in the better ear, HTL asymmetry, and hearing-aid use.**

| Age [a] (years) | | Prevalence of hearing loss % (95%- Confidence Interval) | | Asymmetry [b] % | Hearing-aid adoption rate [c] % |
|---|---|---|---|---|---|
| | | PTA> 25 dB HL | PTA ≥ 35 | | |
| < 55 | Female | 2.4 (1.4–3.7) | 1.2 (0.6–2.3) | 4.9 (3.5–6.7) | 1.0 (0.5–2.0) |
| | Male | 3.0 (1.8–4.8) | 1.2 (0.5–2.5) | 5.4 (3.7–7.7) | 1.7 (0.8–3.1) |
| | Overall | 2.7 (1.9–3.8) | 1.2 (0.7–2.0) | 5.2 (4.0–6.6) | 1.4 (0.8–2.2) |
| 60 | Female | 9.6 (6.7–13.2) | 3.2 (1.7–5.6) | 5.2 (3.1–8.0) | 2.0 (0.8–3.9) |
| | Male | 17.5 (13.1–22.6) | 6.1 (3.6–9.6) | 6.3 (3.8–9.9) | 3.4 (1.6–6.2) |
| | Overall | 13.7 (11.0–16.8) | 4.6 (3.1–6.6) | 5.6 (3.9–7.8) | 2.9 (1.7–4.6) |
| 70 | Female | 29.5 (24.1–35.4) | 14.3 (10.4–19.1) | 12.1 (8.5–16.6) | 8.7 (5.7–12.7) |
| | Male | 35.7 (29.6–42.2) | 15.9 (11.5–21.2) | 14.0 (9.9–19.1) | 10.0 (6.5–14.5) |
| | Overall | 32.4 (28.0–36.9) | 14.7 (11.6–18.4) | 13.4 (10.4–16.9) | 9.1 (6.6–12.1) |
| 80 | Female | 54.1 (46.5–61.5) | 33.9 (27.1–41.3) | 13.7 (9.1–19.5) | 16.3 (11.3–22.4) |
| | Male | 67.9 (61.0–74.2) | 41.7 (34.8–48.8) | 19.2 (14.0–25.2) | 25.4 (19.6–31.9) |
| | Overall | 59.2 (53.6–64.7) | 38.2 (32.9–43.8) | 14.6 (10.9–18.9) | 20.4 (16.1–25.2) |
| 18–97 | Female | 13.8 (12.0–15.8) | 7.9 (6.6–9.5) | 7.0 (5.7–8.4) | 5.1 (4.0–6.4) |
| | Male | 17.4 (15.2–19.8) | 9.3 (7.7–11.2) | 8.4 (6.8–10.2) | 6.1 (4.8–7.7) |
| | Overall | 15.5 (14.0–17.0) | 8.6 (7.5–9.8) | 7.6 (6.6–8.8) | 5.6 (4.7–6.6) |

Weights were applied for overall estimates to adjust age, gender, and educational level to population statistics. Gender-specific prevalence estimates assumed an equal age distribution.

[a] Results refer to age bands 55 to 64, 65 to 74 etc.

[b] Absolute interaural difference in PTA of more than 10 dB.

[c] Uni- and bilateral use of hearing aids, including one user of cochlear implants (female, < 55 years)

The prevalence of $HI_{GBD}$ is slightly lower in our sample than reported for the Dutch sample. Homans et al. reported a $HI_{GBD}$ prevalence of 31% in females and 33% in males aged 65 and above, whereas the age-adjusted estimate for the German data was 27.4% (CI: 22.9–32.4) in females and 30.8% (CI: 24.6–35.4) in males. Compared to US results based on NHANES data, the prevalence of $HI_{WHO}$ is considerably lower in middle-aged and elderly male adults in our study. The estimated 95%-CI for $HI_{WHO}$ by age decade were broad and often close to 10% and even larger. Hence, CI from the US- and the German estimates overlap for most, though not all, age groups. As stated above in the methods section, the differences between the US- and the German estimates can be quantified more conclusively when adjusting the age- and gender distribution of the German 20 to 69 year subsample to the age distribution of the respective US-American survey. Table 3 shows the estimates reported by Agrawal et al. for the NHANES cycles 1999 to 2004 [25] and by Hoffman et al. for the cycles 2011 to 2012 [26], together with the estimates derived in the German subsample applying US sample weights for age.

Interestingly, these simulations yielded basically the same $HI_{WHO}$ estimates for females as published for the NHANES data (approx. 5%) for the 1999 to 2004 and the 2011 to 2012 cycles. Prevalence figures and risks for HI in males, however, were considerably higher in the US data than in the German data, when US sample weights for age were applied. As already expected from the comparison of HTL data, the prevalence of $HI_{WHO}$ in Swedish 70 year- old females was considerably lower than in our 65 to 74 year age groups. Hoff et al. reported a prevalence of $HI_{WHO}$ of 22.2% in females and 27.8% in males [6], whereas we found 29.5% and 35.7%, respectively.

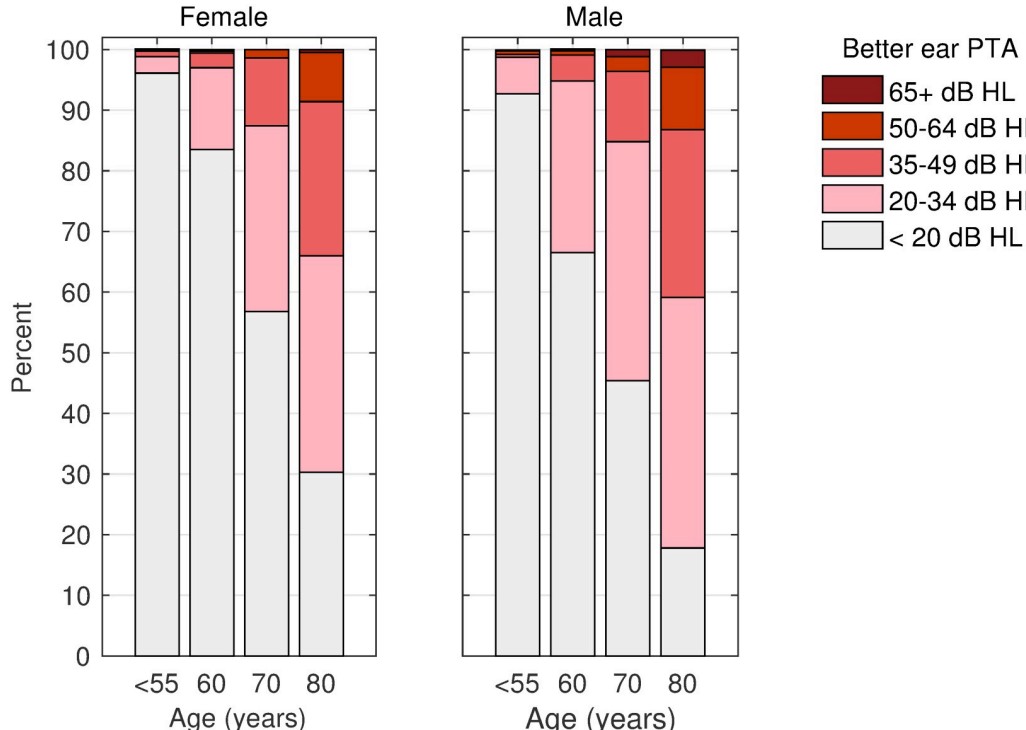

**Fig 1. Degree of hearing loss.** Hearing loss according to the GBD classification [13] in females (left panel) and males (right panel). Categories defining severe to profound HI and profound HI were combined. Sample weights applied to adjust professional education to population statistics.

## PTA increase by age

Comparing prevalence rates across studies, it is essential to bear in mind that minor differences in PTA result in major differences in prevalence numbers when a large proportion of

**Table 3. Prevalence and risk for HI$_{WHO}$ and HI$_{HighFreq}$ in females and males aged 20 to 69 as reported for the US-American NHANES data and as replicated in the German sample.**

| Data base | | | Prevalence | | Odds Ratio (adjusted for age) | |
|---|---|---|---|---|---|---|
| | | | HI$_{WHO}$ | HI$_{HighFreq}$ | HI$_{WHO}$ | HI$_{HighFreq}$ |
| NHANES 1999–2004 | Female | | 4.9 (3.9–5.9) | 10.0 (9.0–12.0) | - | - |
| | Male | | 11.0 (9.0–13.0) | 28.0 (25.0–31.0) | - | - |
| German data (age weights of NHANES 1999–2004) | Female | | 5.0 (3.8–6.5) | 10.9 (9.1–13.0) | - | - |
| | Male | | 7.3 (5.7–9.2) | 21.5 (18.8–24.4) | - | - |
| NHANES 2011–2012 | Female | | 5.1 (3.6–7.3) | 10.6 (8.6–13.0) | 1 | 1 |
| | Male | | 9.9 (7.2–13.3) | 27.6 (23.0–32.8) | 2.4 (1.4–3.9) | 4.8 (3.3.-6.9) |
| German data (age weights of NHANES 2011–2012) | Female | | 5.0 (3.8–6.4) | 11.1 (9.4–13.1) | 1 | 1 |
| | Male | | 7.5 (5.9–9.4) | 22.9 (20.3–25.7) | 1.6 (1.1–2.3) | 3.0 (2.3–3.9) |

Sample weights were applied to the German data to adjust the distribution of age to the samples of the US reports. Based on the data from NHANES 1999 to 2004 cycles [25], OR were estimated in a multivariate model that was not replicable for comparison in our data. Results for the NHANES 2011–2012 cycles were depicted from [5].

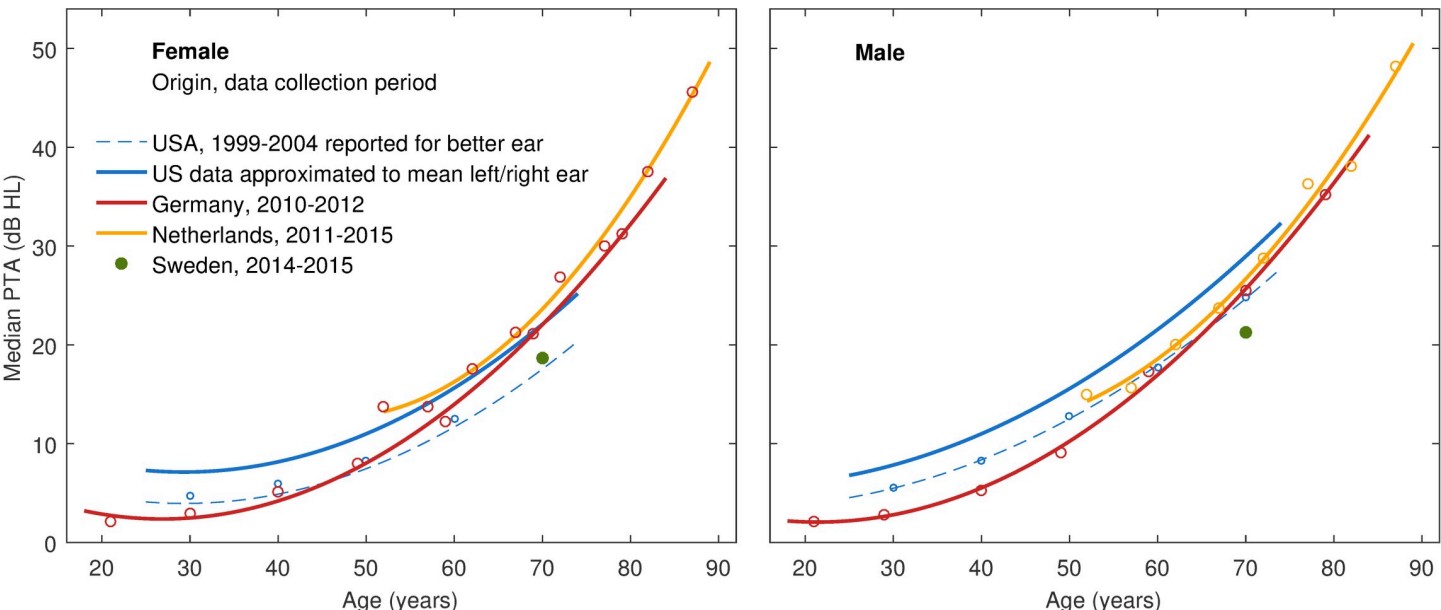

**Fig 2. PTA as a function of age.** Comparison of median PTA from averaged left and right ear in the German sample to published results from international studies for females (left panel) and males (right panel). Median HTL reported for the US study referred to the better ear on a frequency-by-frequency basis and was adjusted using Formula (1).

participants has a PTA around the prevalence cut-off value. Cumulative functions for the elderly age groups are steep in the range of 20 to 30 dB. An increase of only 1 dB at PTA median in the 70 year age group, for example, resulted in an increase of 3 to 4% in HI$_{WHO}$ in our data. Hence slight deviations caused, e.g., by calibration, equipment or measurement conditions can have a considerable impact. To give a better overview of our data and the studies in comparison, the increase of median PTA averaged from left and right ear PTA with age is shown in Fig 2, separately for females and males.

As can be seen, the studies' estimates differ less in females than in males. Consistent with the prevalence rates stated above, the PTA increase by age was lower in our study than in both the US and the Dutch studies, but substantially higher than in the Swedish survey. Particularly pronounced differences were found between the NHANES and our data in young and middle-aged adults. Technical specifics, measurement conditions, and audiometric procedures possibly contributed to the unexpected elevated HTL estimates in the US data, and might have introduced some bias to prevalence estimates using the HI$_{WHO}$ criterion. The adjustment of the US results for the purpose of this comparison, which is also presented in Fig 2, certainly constitutes an additional source of uncertainty, but cannot fully explain these differences.

## Within-study gender gap

Focusing within-study gender group differences minimizes possible technical or methodical inconsistencies that usually arise when comparing results from various studies. The scatter plot in Fig 3 displays the HTL differences between females and males observed in our data in relation to the HTL gender differences in the US, Dutch, and Swedish studies. Note that results for these international studies were reported using different descriptive parameters and different age categories, such that the analysis of our data needed to be adapted separately to each study. Symbols above the diagonal line indicate that the gender gap is smaller in our data than in the comparative data. Conversely, markers below the diagonal indicate a larger gender gap

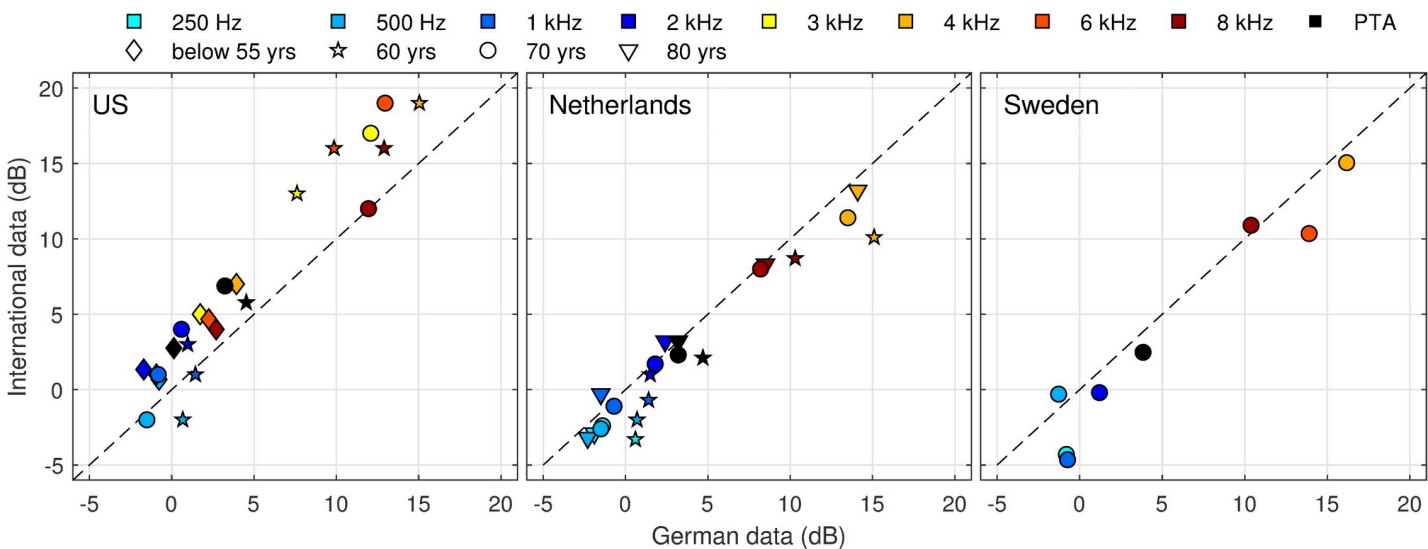

**Fig 3. HTL gender gap observed in four population-based studies.** HTL differences were calculated subtracting mean/median HTL derived in females and males by frequency and age group (male minus female). Note: The analysis of the German data was aligned separately to the methods used in each study. Comparison was performed for the better ear on a frequency-by-frequency basis (US) reported in [4,26], mean right and left ear HTL averaged over two 5-years bands (NL) reported in [10], and interpolated/grouped median (SE) reported in [6].

in our data. In the German study, HTL gender gaps were well below 5 dB at all frequencies in young to middle-aged adults. They were equally narrow at frequencies up to 2 kHz for the 60 to 80 year age bands. At frequencies above 2 kHz, however, HTL gender gaps varied between 7 and 16 dB depending on age, frequency, and descriptive parameter. Whereas females were favored at high frequencies, the HTL gap was often reversed at frequencies below 2 kHz.

Overall, the HTL gender gaps in our study were well in line with the Swedish and the Dutch data, except for the 60 year age group (stars). The gender gap was larger in 60 year-old German participants than in Dutch adults. In contrast, HTL gender gaps were almost consistently broader in the NHANES results than in our data. These gap differences between both studies were smaller for young- to middle-aged adults, but markedly larger for 60 and 70 year adults. This comparatively large gender gap at the high frequencies HTL observed for the 1999 to 2006 NHANES cycles was basically confirmed in the latest 2011 to 2012 cycles. The prevalence estimates for $HI_{highFreq}$ reported for males and females aged 20–69 differ by 18% in the 1999 to 2004 cycles and 17% in the 2011 to 2012 cycles (Table 3). In our data, however, the gender gap for $HI_{highFreq}$ was only 11% and 12% when sample weights were applied to achieve the age distribution of the respective NHANES cycles. Accordingly, the age-adjusted risk for $HI_{highFreq}$ was higher in US-American males (OR 4.8) than in the German sample (OR 3.0).

## Discussion

The present analysis is based on HTL data derived from 3105 adults aged 18–97 from two independent studies that were conducted between 2010 and 2012 in three different regions of Germany. The distribution of HTL by age and gender, as well as the prevalence of HI using two competing criteria, were estimated and compared to results derived from other European societies and the US.

Prevalence according to the WHO and the GBD criteria was 15.5% and 8.6%, respectively, with sample weights applied to adjust age, gender and educational level to population statistics for the year 2011. The proportion of hearing-aid adoption was 5.6%, thus lagged considerably

behind either prevalence estimate, an aspect that is addressed in detail with regard to pure-tone and speech audiometric results as well as to subjective hearing difficulties elsewhere [27]. Homans et al. used population weights for the year 2015 and generalized that 33% of the men and 31% of the women above the age of 65 years have a hearing loss of 35 dB or more—in the Netherlands and presumably in other European countries [10]. In the present analysis, the year of reference for the population weights was 2011, resulting in an age-adjusted $HI_{GBD}$ prevalence of 27.4% in women and 30.8% in men aged 65 and above. Using the year 2015 as the reference for the German data [28,29], would increase $HI_{GBD}$ estimates to 28.1% in females and 32.2% in males aged 65 and above. Taking the uncertainties of prevalence estimation and the substantial impact of demographic characteristics—particularly the increasing proportion of adults aged above 80—into account, our numbers are similar to the findings of the Rotterdam study. Hence, the assumption of Homans et al. that the prevalence number of the Dutch sample is "representative for most populations in other Western European countries" [10] basically holds for the German population.

### Comparison of age-related hearing loss and gender differences

There was no common ground to compare the most recent results from Europe and the US. Statistical approaches vary, and standard populations like the WHO or the EU standard populations to facilitate comparison are underused in hearing research. Therefore, alignments were made and specific values from various statistical approaches were combined that allow at least indirect conclusions to be drawn based on the HTL data collected within a rather tight time frame from 2008 to 2015. From a general point of view, the HTL increase with age and the prevalence of HI seems less pronounced in the current European data than in US data. This can be traced back to considerably better HTL and, subsequently, lower HI prevalence in males, since the findings for females were very similar in the US-American, the Dutch and the German studies—except the Swedish study which reported the least HTL increase with age for both genders. Accordingly, the gender gap in HTL and HI prevalence increase by age is much less pronounced in recent European studies than in the US results–and in earlier European studies. These findings provide further evidence to the hypothesis that modifiable factors might be involved in the etiology of HI across lifespan to a considerable extent.

In a previous analysis, age-related HTL and $HI_{WHO}$ prevalence estimated from the HÖRSTAT sample were compared to a number of population-based studies conducted earlier in Europa, Australia, and the US [8,9]. The increase of HTL and $HI_{WHO}$ prevalence with age was shown to be lower in the unweighted HÖRSTAT sample than reported for US-American studies: Beaver Dam [12] conducted 1993 to1995, Beaver Dam Offspring [30] conducted 2005 to 2008, various NHANES cycles and–though not included in [8,9]–the Framingham Cohort [11], conducted 1983 to 1985. Varying in degree, this also applied to the very large Norwegian HUNT II study [31,32] conducted 1996 to 1998, the noise-screened Swedish Östergötland sample [33] conducted 1999 to 2000, the Australian SAHOS study [34] conducted presumably in the 1990s, an Italian multi-center study [35] conducted presumably in the early 1990s, and the British MRC study [36] conducted 1980 to 1986. Only the Australian Blue Mountains Hearing study, conducted 1997 to 2000, gave comparable results [37]. If results were reported separately for females and males, the gender gaps in HTL and HI prevalence were larger in all other studies than in HÖRSTAT, with the exception of the noise-screened Östergötland sample. Owing to the good agreement of the Aalen and the HÖRSTAT results, these findings were overall confirmed when replicating these multiple comparisons in the Aalen data set and the pooled data set [14]. Measurement procedures (automated vs clinical audiometry with experienced personnel), conditions (sound-booth vs places largely uncontrolled for ambient noise),

and equipment are important influencing factors that principally complicate the comparison of prevalence estimates from different studies. In one instance or another, these factors might provide some explanation as sampling and subject recruiting do. Additionally, the period of data collection needs to be considered. Various aspects that support a postponed onset of hearing loss were discussed in [8] and entail noise-abatement law, tertiarization of employment, expanded health prevention, and improved medical management of risk factors such as diabetes. In particular, a changed work environment possibly had a greater impact in male-dominated occupational areas. These factors, easier to invoke in a grand gesture than to disentangle in their effects, apply similarly to other western societies. Hoff et al. [6], as well as Zhan et al. [3], observed stronger birth cohort effects in males than in females. Hoffman et al. found about the same relative decline of the $HI_{WHO}$ prevalence in men and women [5], which still implies that the absolute decline in HI prevalence was stronger in males than in females. Since the marked differences between European and US-American estimates for males seem insufficiently explained by the abovementioned factors, other possible reasons may be speculated upon. The health systems of the countries included in the comparison differ markedly. More importantly, there is a general difference between the European countries and the US. The European countries essentially pursue systems regulated by the state or by social partnerships and often sustained by compulsory insurances, which in principle guarantees health care for every citizen. In contrast, health care is mostly based on private arrangements in the US and governmental health care regulations reach only a comparatively small proportion of the population. According to the official statistics for the year 2011, 17.7% of the US citizen aged 18 and more years had no health insurance coverage at any time during the year [38]. As showed by Freeman et al. (2008) in their systematic review, however, health insurance had substantial effects on the use of health services and health outcomes [39]. By this means, widespread pathologies that might impact hearing may remain less frequently undiagnosed and receive more frequently continuous medical treatment in Europe than in the US. For example, there is evidence that males are overall at a higher risk for diabetic diseases [40] and that diabetic diseases should be regarded as independent risk factor for hearing loss [41,42]. Thus, systemic undercoverage in health care could have indirectly contributed to the comparatively large gender gap in the US results. Moreover, the exposure to firearm noise in military service and in leisure time might also contribute to explaining this transatlantic difference. The vast majority of German male study participants aged 80 had either never served in combat or had done, if any, only basic military service, because the professional army was comparatively small. This holds, overall, for most European societies, but not for the adult US-American society, which faced several military operations after World War II and included almost 10% of veterans in the 2015 adult population [43]. However, it is questionable whether the higher proportion of men who have done military service contributed to a higher HI prevalence in US males. Wilson et al. (2010) used various measures for comparing the hearing abilities and performance of 999 male veterans and 590 male non-veterans in the EHLS data [44]. They found veterans and non-veterans equally likely to have a hearing loss. Nevertheless, the use of firearms is heavily regulated in European countries and, in any case, much less popular than in the US [45]. For these reasons, it can be assumed that HI due to firearm noise and shooting is not a broadly-based societal issue in Europe. Military commitment and private firearm use possibly contribute to the comparatively large gender gap in $HI_{WHO}$ and $HI_{highFreq}$, but cannot sufficiently explain the differences between the studies. In the US data, male gender was associated with a notably higher risk for HI than in our data. Even if a great variety of factors like age, noise exposure including firearms, educational level, and cardiovascular risk factors was controlled for in multivariate analyses, OR in males were 1.8 for $HI_{WHO}$ and 3.8 for $HI_{highFreq}$ in the most

current NHANES data [5]. Thus they were still higher than the estimates in our data, which resulted in OR = 1.6 and OR = 3.0 when adjusting for age only.

Although the European studies point in the same direction when compared to the US results, it should be noted that our and the Dutch estimates for adults aged 70 differ considerably from the Swedish estimates. The fact that a group including only one birth year was not strictly comparable to a group including 65 to 74 year-old adults does not provide a fully plausible explanation. The Swedish [6] and the Dutch [10] had a similarly high response rate, about 70 and 80%, but were single-center studies. Additionally, the audiometric procedure was different in the comparison studies. In the Dutch and in our study, HTL were measured manually using an ascending procedure, whereas the Swedish study applied a combined descending and ascending procedure (like NHANES). More importantly, the maximum output level was limited to 90 dB HL in the Gothenburg study and no information is given as to how out-of-range HTL were considered in the analysis. In the Dutch and in our study, by contrast, the test signals met the requirements for conventional clinical measurements, with output levels above 110 dB HL in the mid frequencies. Out-of-range HTL were noted as the maximum output level plus 5 dB. If data were excluded in the case that no response was given by the subject in the Gothenburg study, this, together with the low output level limit, might to some degree explain the notably better HTL in the Swedish data. Otherwise, the question remains open as to how to interpret these intra-European deviations, apart from a vague reference to the long-standing and particularly strong social welfare system in Sweden.

## Gap reversal in the low frequencies

Consistent with previous findings, in the present study HTL at low frequencies up to 1 kHz were poorer in women than in men, e.g. [32,33,46,47]. This reverse gender gap increased with age. In terms of dB, the difference was notably smaller than the HTL gap in the high frequencies and consequently cannot "compensate" for the HTL difference at 4 kHz when using PTA. However, this reversed gender gap was more pronounced in the Dutch and the Swedish (about 2–5 dB below 2 kHz) than in the US and our study. Among the hypotheses to explain the gap reversal in the low frequencies, the atrophy of the stria vascularis has long been discussed. Schuhknecht described the phenomenon and termed it "metabolic presbycusis" [48]. More recently, Dubno and collegues included a metabolic type in their classification of audiometric phenotypes; this is characterized by a gently sloping HTL in the high frequencies and comparably poor HTL in the low frequencies [49,50]. Thus the progression of hearing loss in females and males differs not only in relation to the temporal pattern, but might also disagree in its spectral pattern and type.

## Prevalence, demographics and absolute numbers

Reporting age-adjusted prevalence figures for males and females includes the fact that HI was mainly a female issue in Germany. In 2011, HI with PTA $\geq$ 35 dB affected approximately 3.2 million adult females vs 2.6 million adult males, simply because women accounted for a larger share in the general elderly population. Owing to their higher life expectancy, the mean age of females was–and still is–higher than the mean age of males. This is true for all European- and for the US population. Among the adults aged 65 and over in 2015, the percentage of women was 54% in Sweden [51], 55% in the Netherlands [52], 57% in Germany [28], and 56% in the US [53], and the female share sharply increases from decade to decade. Thus, looking at the absolute numbers, the gender issue in hearing loss is exactly the opposite to that usually assumed.

## Strengths and Limitations

One strength of the present analysis is that it made use of a pooled database from two population-based studies that were carried out in three geographically distinct areas. This data base and the size of the subsamples were assumed to be large enough to reliably estimate HTL percentiles separately by gender for age groups between 20 and 80. But still, there are three limitations of this analysis. First, the response rates were low in both studies compared with international studies. However, if the design as singular cross-sectional hearing studies, as well as specific conditions like the fragmented health system, lack of monetary compensation of expenses, and mentalities were taken into account, it could be considered to be satisfactory. All the more since in this part of the world the abundance of surveys and data collections meanwhile arouses more suspicion than enthusiastic participation. Against this background, similar low and even lower response rates were reached in the pretest studies for the German National Cohort, a large-scale health study, despite monetary incentives and additional recruitment approaches [54, 55]. Low response rates need to be discussed critically, but without overvaluing this single indicator regards to potential biases in the studies' results. Low response rates do not necessarily lead to biased response, just as biased response cannot be excluded when the response rate is high [56]. Secondly, socially least-advantaged strata and presumably adults with non-German primary language were underrepresented in the samples. Whilst the former was arguably balanced out by adjusting for professional education, the low percentage of participants with first languages other than German (5% in the HÖRSTAT study sample) questions whether the sample was representative for the general population on ethnic lines. However, ethnic affiliations were not recorded neither in the present study or any other earlier hearing study in Germany. Hence, there is no basis for ethnicity-specific comparisons to analyses of US data where hearing loss was found to be more pronounced in white Americans than, particularly, in black Americans of the same gender and age [25, 57]. However, this finding was not confirmed by Dawes et al. (2014) for the large UK Biobank database [58]. In contrast to the US results, odds for hearing impairment measured with a speech recognition test in noise increased with ethnic minority backgrounds, including British black. Thirdly, estimates for age cohorts above the mean life expectancy are critical and per se subject to a high degree of uncertainty. Poor health and increasing multimorbidity is certainly more prevalent in higher-aged cohorts and probably restrained participation both in the HÖRSTAT and the Aalen study. Accordingly, a kind of healthy-enough-bias that probably led to underestimating hearing loss in high-aged adults cannot be ruled out completely. Since the prevalence of HI was high in these cohorts whichever criterion was applied, the overall prevalence estimates for adults aged 18+ are presumably affected, albeit possibly to a minor degree.

## Conclusion

The analysis of pooled data from two population-based German studies yielded a lower HTL increase by age and a lower risk for HI in male adults than reported in US-American surveys. Compared to European studies which were conducted at about the same time, a good agreement of HI estimates was found to a large Dutch study. This agreement was basically not confirmed with regard to a large Swedish study which resulted in lower HTL estimates. Overall, these partly pronounced differences derived in recent epidemiological surveys of pure-tone hearing support retaining the recommendation to compile diverse national data sets when revising ISO 1999. More interestingly, and almost irrespective of the HTL increase by age, the gender differences in pure-tone hearing were markedly smaller in the European studies than in the current US results. This observation might reflect profound societal changes in Europe

that possibly tend to balance out the unequal distribution of risk factors for pure-tone hearing between genders.

## Acknowledgments

The authors thank Nienke C. Homans, who kindly provided HTL percentile data for the results published in [10]. Language services were provided by www.stels-ol.de.

## Author Contributions

**Conceptualization:** Petra von Gablenz, Eckhard Hoffmann, Inga Holube.

**Data curation:** Petra von Gablenz.

**Formal analysis:** Petra von Gablenz.

**Funding acquisition:** Eckhard Hoffmann, Inga Holube.

**Methodology:** Petra von Gablenz, Inga Holube.

**Project administration:** Petra von Gablenz.

**Supervision:** Eckhard Hoffmann, Inga Holube.

**Validation:** Petra von Gablenz.

**Visualization:** Petra von Gablenz.

**Writing – original draft:** Petra von Gablenz.

**Writing – review & editing:** Petra von Gablenz, Eckhard Hoffmann, Inga Holube.

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
