## [Decision Letter · Decision Letter 0]

5 Feb 2020

PONE-D-19-25992

Gender-specific hearing loss in German adults aged 18 to 84 years compared to US-American and current European studies

PLOS ONE

Dear Dr. von Gablenz,

Thank you for submitting your manuscript to PLOS ONE. After careful consideration, we feel that it has merit but does not fully meet PLOS ONE’s publication criteria as it currently stands. Therefore, we invite you to submit a revised version of the manuscript that addresses the points raised during the review process.

Please follow the reviewer's comments in revising the manuscript.

We would appreciate receiving your revised manuscript by Mar 19 2020 11:59PM. To enhance the reproducibility of your results, we recommend that if applicable you deposit your laboratory protocols in protocols.io, where a protocol can be assigned its own identifier (DOI) such that it can be cited independently in the future. For instructions see: http://journals.plos.org/plosone/s/submission-guidelines#loc-laboratory-protocols

We look forward to receiving your revised manuscript.

Kind regards,

Nayu Ikeda, Ph.D.

Academic Editor

PLOS ONE

Journal Requirements:

Reviewers' comments:

Reviewer's Responses to Questions

**Comments to the Author**

1. Is the manuscript technically sound, and do the data support the conclusions?

Reviewer #1: Yes

2. Has the statistical analysis been performed appropriately and rigorously? 

Reviewer #1: Yes

3. Have the authors made all data underlying the findings in their manuscript fully available?

Reviewer #1: Yes

4. Is the manuscript presented in an intelligible fashion and written in standard English?

Reviewer #1: Yes

5. Review Comments to the Author

Reviewer #1: The authors compared gender-specific hearing loss in German adults aged 18 to 84 years compared to US and European studies. I have agreed to review the paper as I have previously published in this area:

Scholes, S., Biddulph, J., Davis, A., & Mindell, J. S. (2018). Socioeconomic differences in hearing among middle-aged and older adults: cross-sectional analyses using the Health Survey for England. BMJ open, 8(2), e019615.

However, I am not a technical hearing expert: so cannot comment on the various technical aspects of the study, especially the hearing threshold level (HTL) aspect of the study. The paper is very well written and I applaud the authors for their efforts in making detailed comparisons across the different studies. I only have a number of points that the authors way wish to consider.

Abstract

Rather than refer to “international standards such as ISO 1999” which readers such as myself may be unfamiliar with could the authors point out the direction of the gender difference: rather than simply saying that the gender gap was smaller in Germany compared to the US.

Introduction

The authors should give the study location for the study by Zhan et al (REF 3) to match the details given for the other studies. Given the reference to gender gap I was surprised that the authors did not refer to this in their research hypotheses.

Methodology

I did not understand the sentence (line 206-207) which states that “Generally, measurement data were regarded as grouped data”. The authors should state clearly what they mean without readers having to look-up the reference. I also found it very surprising that the authors relied on non-overlapping confidence intervals to consider between group differences as statistically significant. This is a conservative approach: would a formal test have been better? In the results the authors state that the “gender differences failed statistical significance…” is this because of the use of non-overlapping CIs?

Results

The authors refer to the mean for the results presented in Table 1: this is confusing. I assume they meant medians. Perhaps frequency in Table 1 would be better labelled as referred to in the Methods section: i.e. 0.25, 0.5, 1, 2, 3, 4, 6, and 8

Discussion

I applaud the wishes of the authors to avoid making ‘vague references’ to factors such as social welfare provision for differences in hearing levels between countries. However I was wondering whether the comparisons between the US and German estimates would be strengthened by a consideration of race/ethnicity for the former. The authors briefly mention ethnicity as a limitation of their study. However, I would have liked a little more on the discussion of ethnicity. Have there been any German studies that shed light on ethnic variations in hearing? I also wonder whether the discussion is short on policy and public-health implications. Although age-related hearing impairment seems to be less smaller in Europe what initiatives are – and could be put – in place to continue to reduce age-related hearing loss. For example, although the authors present estimates for hearing aid adoption they do not discuss who pays for the costs associated with hearing aid use?

Figures

I would like the figures to have higher resolution in a final version: plus a consideration of using black and white rather than colour.

But clearly this is an impressive study with which I have little to object against!

6. PLOS authors have the option to publish the peer review history of their article (what does this mean?). If published, this will include your full peer review and any attached files.

Reviewer #1: Yes: Dr Shaun Scholes

---

## [Author Response · Author response to Decision Letter 0]

20 Mar 2020

Dear Dr Scholes,

thank you very much for your constructive and encouraging comments on the manuscript that clearly helped to improve the manuscript. We've uploaded a separate file including your comments and our responses.

Best regards,

Petra von Gablenz

---

## [Editor Report · Decision Letter 1]

30 Mar 2020

Gender-specific hearing loss in German adults aged 18 to 84 years compared to US-American and current European studies

PONE-D-19-25992R1

Dear Dr. von Gablenz,

We are pleased to inform you that your manuscript has been judged scientifically suitable for publication and will be formally accepted for publication once it complies with all outstanding technical requirements.

With kind regards,

Nayu Ikeda, Ph.D.

Academic Editor

PLOS ONE
---

## [Editor Report · Acceptance letter]

9 Apr 2020

PONE-D-19-25992R1 

Gender-specific hearing loss in German adults aged 18 to 84 years compared to US-American and current European studies 

Dear Dr. von Gablenz:

I am pleased to inform you that your manuscript has been deemed suitable for publication in PLOS ONE. Congratulations! Your manuscript is now with our production department. 

With kind regards,

on behalf of

Dr. Nayu Ikeda 

Academic Editor

PLOS ONE